# Effects of Different Exercise Types on Chrna7 and Chrfam7a Expression in Healthy Normal Weight and Overweight Type 2 Diabetic Adults

**DOI:** 10.3390/biomedicines11020565

**Published:** 2023-02-15

**Authors:** Keryma Chaves da Silva Mateus, Ivan Luiz Padilha Bonfante, Amanda Veiga Sardeli, Renata Garbellini Duft, Arthur Fernandes Gáspari, Joice Cristina dos Santos Trombeta, Joseane Morari, Bruno Rodrigues, Márcio Alberto Torsoni, Mara Patrícia Traina Chacon-Mikahil, Licio Augusto Velloso, Cláudia Regina Cavaglieri

**Affiliations:** 1Exercise Physiology Laboratory, Faculty of Physical Education, University of Campinas, Campinas 13083-851, SP, Brazil; 2Federal Institute of Education, Science and Technology of São Paulo, Hortolândia Campus, Hortolândia 13183-250, SP, Brazil; 3Gerontology Program, Faculty of Medical Sciences, University of Campinas, Campinas 13083-887, SP, Brazil; 4Institute of inflammation and Ageing, University of Birmingham, Birmingham B15 2TT, UK; 5Faculty of Applied Sciences, University of Campinas, Limeira 13484-350, SP, Brazil; 6Obesity and Comorbidities Research Center, University of Campinas, Campinas 13083-864, SP, Brazil

**Keywords:** obesity, type 2 diabetes mellitus, CHRNA7, CHRFAM7A, physical exercise

## Abstract

**Purpose:** Considering that the CHRNA7 and CHRFAM7A genes can be modulated by acute or chronic inflammation, and exercise modulates inflammatory responses, the question that arises is whether physical exercise could exert any effect on the expression of these genes. Thus, the aim of this work is to identify the effects of different types of exercises on the expression of the *CHRNA7*, *CHRFAM7A* and tumor necrosis factor-*α* (TNF-*α*) in leukocytes of healthy normal weight (HNW), and overweight with type 2 diabetes (OT2D) individuals. **Methods:** 15 OT2D and 13 HNW participants (men and women, from 40 to 60 years old) performed in a randomized crossover design three exercise sessions: aerobic exercise (AE), resistance exercise (RE) and combined exercise (CE). Blood samples were collected at rest and post-60-min of the exercise sessions. The leukocytes were the analysis of the *CHRNA7*, *CHRFAM7A* and (*TNF-α)* gene expression. **Results**: At baseline, OT2D had higher *CHRFAM7A* and *TNF-α* expression compared to HNW. No statistical differences were observed between groups for *CHRNA7*; however, the HNW group presented almost twice as many subjects with the expression of this gene (24% vs. 49%). Post exercise, the *CHRFAM7A* increased in AE, RE and CE for HNW, and in AE and CE for OT2D. There was no significant difference for *TNF-α* and CHRNA7 expression between any type of exercise and group. **Conclusions:** Our study shows that OT2D individuals presented higher baseline expression of TNF-α and CHRFAM7A, besides evidence of decreased CHRNA7A expression in leukocytes when compared with HNW. On the other hand, acutely physical exercise induces increased CHRFAM7A expression, especially when the aerobic component is present.

## 1. Introduction

Since 1975, the percentage of obese individuals has tripled around the world [1]. Nowadays, it is estimated that more than 2.5 billion people are overweight and/or obese, which characterizes a serious pandemic [1]. Obesity is associated with the development of noncommunicable diseases, such as type 2 diabetes mellitus (T2DM) [2]. Among the physiological mechanisms connecting obesity and T2DM is the increase in chronic low-grade inflammation [3,4].

There is evidence for an association between alterations in activation of the α7 subunit of the nicotinic receptor (α7nAChR) in peripheral tissues and increased local inflammation and metabolic disorders [5,6,7]. Despite it not being completely understood, this pathway comprises neural and non-neural crosstalk between the central nervous system and lymphocytes. The higher inflammatory state leads to acetylcholine production via T lymphocytes [8,9], which in turn binds the α7nAChR at the periphery and inhibits the pro-inflammatory responses, which has being called the cholinergic anti-inflammatory pathway (CAP) [10]. Some evidence shows that in animals and humans there is an association between alterations in α7nAChR expression of peripheral tissues and increased local inflammation and metabolic disorders [5,6,7].

The α7nAChR is expressed by the cholinergic receptor nicotinic alpha 7 subunit *(CHRNA7)* gene in some cells, such as leukocytes [11,12,13,14]. These cells also appear to present the partially duplicated *CHRNA7,* the cholinergic receptor, nicotinic, and alpha 7 family with sequence similarity 7A *(CHRFAM7A)* gene, which has been demonstrated to act as a dominant negative regulator of *CHRNA7* in human and in vitro studies [11,14,15]. It is expected that inflammatory agents may be able to modulate the expression of both genes [10,13].

In injury and infection situations such as COVID-19 [16], brain ischemia [17] and skin burn [18], the downregulation of *CHRFAM7A* expression is associated with a worse prognosis. This counterregulatory effect appears to be an important mechanism for acute infection and injury control [11,14]. On the other hand, in chronic inflammatory baseline conditions such as inflammatory bowel disease, this control mechanism undergoes alteration through a *CHRNA7* reduction and the upregulation of the *CHRFAM7A*, further increasing inflammation [11,14,15,19]. It is still unknown whether the gene expression response of both genes against a stressor agent would be similar in subjects who are already onto chronic inflammatory conditions such as obesity and T2DM, compared to what is known for healthy individuals.

Physical exercise can trigger an inflammatory response immediately after the session [20]. The nature of the response, being either more pro or anti-inflammatory, depends on the characteristics of the exercise protocol, such as the metabolic demand during exercise, and the magnitude of muscle damage [21]. Typically, a resistance exercise bout would promote mechanical stress and muscle damage in previously untrained individuals, while an aerobic exercise bout can induce greater oxidative stress [22]. Considering that the CHRNA7 and CHRFAM7A genes can be modulated by acute or chronic inflammation, the question that arises is whether physical exercise could exert any effect on the expression of these genes. Cancello et al. (2012) demonstrated that the change in lifestyle with the inclusion of regular physical exercise, for 3 months, increased the expression of α7nAChR in the adipose cells of obese subjects together with the improvement of inflammatory parameters [23]. The acute effect of each exercise bout is believed to be one of the determinant mediators of chronic training adaptations, namely, the release of cytokines from the exercising muscles and the mobilization of inflammatory cells (immunosurveillance) by the cortisol and norepinephrine stimuli [21,24]. To our knowledge, there are no studies that have investigated the acute effect of a physical exercise session on the expression of both genes in healthy and/or chronically diseased humans.

Thus, the first aim was to test whether obese individuals with T2DM have CHRNA7 downregulation and CHRFAM7A and TNF-α upregulation expression in leukocytes when compared to healthy individuals without T2DM. The second aim was to identify the acute exercise effects on the gene expression of these genes in leukocytes; due to evidence of the physiological protective factors of CHRFAM7A observed in certain diseases and acute stressful conditions, we hypothesized there will be an increase in the expression of CHRFAM7A after an exercise session, especially in healthy individuals. As exercise effects on gene expression may vary according to the energetic expenditure, sympathetic activation, muscle damage and blood factors such as oxidative stress during the session, we also aim to compare the effects of different types of exercise on the expression of *TNF-α*, *CHRNA7* and *CHRFAM7A* genes in leukocytes of overweight individuals with T2DM (OT2D) and healthy normal weight individuals (HNW).

## 2. Materials and Methods

### 2.1. Subjects

This study is part of a randomized controlled clinical trial (UTN: U1111-1202-1476, http://www.ensaiosclinicos.gov.br/rg/RBR-62n5qn/ accessed on 10 November 2022), which evaluated the effects of exercise training protocols on physical fitness and molecular/metabolic/inflammatory markers in middle-aged, healthy, normal weight people, and overweight people with and without T2DM (for the purpose of the present study, only OT2D and HNW groups were analyzed). This study was approved by Ethics Committee of the University of Campinas. All participants gave written informed consent in accordance with the Declaration of Helsinki.

Participants were invited by advertisements on the university campus and by local media. The interested participants contacted us by telephone or email contact and after responding to structured anamneses, the ones meeting the inclusion criteria were recruited. Inclusion criteria were middle-aged men and women (40 to 60 years old) who had not participated in regular exercise programs for the previous 12 months according to the Baecke Habitual Physical Activity Questionnaire [25]. Participants with body mass index (BMI) 25 to 34.9 kg/m^2^ were considered overweight and T2DM was identified through a medical diagnosis. Healthy normal weight participants were included in this group if they had a BMI between 18.5 and 24.9 kg/m^2^ and no medical diagnosis of T2DM.

Exclusion criteria were coronary artery disease, severe hypertension, chronic obstructive pulmonary disease, limited osteoarticular diseases, to be a smoker or to be under use of any medication that could interfere in the physiological responses of testing or training, such as exogenous insulin, thiazolidinedione, dopaminergic agonists, adrenergic inhibitors and anti-inflammatory medication.

Furthermore, the discontinuity criteria were an absence in one or more acute exercise sessions and noncompliance with the recommendations to do acute physical exercise. Additionally, participants undergoing acute inflammation before any of the test days were excluded and only those with full data available were included in the final analysis.

A total of 393 participants underwent the first screening. Three hundred seven were excluded or disapproved in the clinical examination which was carried out by a cardiologist and included an electrocardiogram during rest and during the effort to ensure the participants’ safety. Thirty-six participants were initially included; however, eight were excluded according to discontinuity criteria. Thus, 28 participants were included: 15 overweight individuals with T2DM (OT2D) and 13 healthy normal weight individuals (HNW) were included (Figure 1).

As mentioned, this study was part of a larger trial and the a priori power calculation was conducted for other variables [26,27]. Therefore, we just presented the a posteriori power calculation for baseline genes and pre-post expression CHRFAM7A in all exercise conditions. The power was calculated using G*Power 3.2.1 software (University of Kiel, Kiel, Germany), using a study design of an f test and/or *t* test with *p* < 0.05 and 95% power. The baseline power results were: CHRN7A t = 0.13; CHRFAM7A t = 0.47; TNF-α t = 0.99. In the exercise sessions, OT2D showed a CHRFAM7A power of: AE f = 0.42, t = 0.20; RE f = 0.18, t = 0.12; CE f = 0.38, t = 0.19, while HNW showed a power of AE f = 0.67, t = 0.29; RE f = 0.36, t = 0.19; CE f = 0.22, t = 0.15.

### 2.2. Study Design

HNW and OT2D groups performed the three acute sessions with a randomized crossover design: aerobic exercise (AE), resistance exercise (RE) and combined exercise (CE). The sessions were separated by a one-week washout period.

Body composition and basal blood samples were assessed in the first visit. On the second day, participants performed the cardiorespiratory test. After 72 h, an adaptation session was conducted with all equipment used in the physical exercise sessions. The 12 maximum repetitions test (12RM) (to identify the ideal load a person can use to perform 12 repetitions, but not 13 or more) was performed as well as a retest on the seventh and tenth days, respectively. After one week of washout, participants underwent the acute physical exercise sessions.

A standard breakfast was provided 120 min before each acute exercise session. It was composed of 20 g Quaker^®^ (São Paulo, Brazil) oats, 90 g light Nestlé^®^ (São Paulo, Brazil) Greek yogurt, a Club Social^®^ (São Paulo, Brazil) wafer pack and a banana (44 g carbohydrates, 10 g protein, and 7 g lipids, a total of 278 calories). To perform the acute physical exercise sessions, participants were instructed to take the same medications, ingest only the standard breakfast given, to abstain from caffeine or any stimulant drink during the previous 12 h, alcohol during the previous 24 h and physical exercise during the previous 72 h. Participants were also instructed to maintain their eating and sleeping habits in the days/nights prior to the three exercise sessions, however this was not controlled.

They arrived at the laboratory between 7:30 a.m. and 9:30 a.m. Before starting the session, a systolic (SBP) and diastolic (DBP) blood pressure were measured using the oscillometric method to ensure their safety. Immediately after, blood glucose measurement was performed using a glucometer device (G-tech free lite^®^, Rio de Janeiro, Brazil), only for the OT2D group with the aim of maintaining similar baseline glycemic levels between physical exercise sessions. Then blood collection was performed before the beginning of the session and after 110 min post exercise session.

The AE, RE and CE were equalized to maintain a duration of 50 min. Another blood sample was collected 110 min after the end of the session. The sequences of assessment’s acute experimental protocols are described in Figure 2.

### 2.3. Body Composition and Physical Fitness Evaluations

The body composition of the participants was estimated via plethysmography using the Bod Pod™ (COSMED USA, Inc., Concord, CA, USA) body composition system.

Maximum cardiorespiratory test followed standard procedures [28], where the gas exchange was collected continuously using an automated breath-by-breath metabolic cart (CPX, Medical Graphics, St. Paul, MN, USA). The mean of the highest 30-s value of oxygen consumption was analyzed as the peak oxygen consumption (VO_2peak_).

The 12-repetition-maximal test, adapted from Abdul-Hameed et al. (2012), was used to determine the load of following exercises that were performed in RE and CE sessions: chest press, leg press 45°, lateral pulldown, leg curl, triceps pulley, leg extension, arm curl, calf press on the leg press 45°. More details on 12-repetition-maximal test are in Bonfante et al. (2022) [26].

### 2.4. Blood Sampling

Approximately, 20 mL of blood was collected from the antecubital into Vacutainer^®^ tubes (Becton Dickinson Ltd., Oxford, UK) for plasma and serum samples, in the morning (07h00–08h00 a.m.), after a 12 h overnight fast and 72 h absence of physical exercise or assessment. Part of the samples was collected and sent to a clinical laboratory to quantify blood glucose, insulin (chemiluminescence method), glycated hemoglobin (A1c) (high-performance liquid chromatography method), and lipid profile concentrations (enzyme-trind method, accelerator-selective detergent method and Friedewald formula). Based on the insulin and glucose values, the HOMA-IR was calculated [29]. Another part of the samples was used to separate blood leukocytes and analyze the *CHRNA7*, *CHRFAM7A* and *TNF-α* expression.

### 2.5. Leucocytes Extraction

In acute sessions, 6 mL of blood samples were collected in Vacutainer^®^ tubes (Becton Dickinson Ltd., Oxford, UK) (containing EDTA) for plasma samples at pre and post 60-min of the recovery period for gene expression analysis. Both blood samples were taken using an adapter plug inserted into the brachial vein at antecubital fossa, which remained throughout the exercise session and the recovery period.

Immediately after the blood collection, leukocytes were separated through the use of a reagent capable of separating cells by density gradient (Histopaque^®^, Sigma, Saint Louis, MO, USA). To do this, we put 4 mL of this reagent in a conical tube, and on it, 6 mL of blood were added gently, previously homogenized. This mixture was then centrifuged for 10 min at 3000 rpm and the upper layer containing the leucocytes was harvested and transferred to a new conical tube. These cells were washed with 0.9% saline and again centrifuged for 10 min at 3000 rpm. The saline solution was discarded, and the resulting leucocyte pellet was added, 2 mL of Trizol^®^ (Invitrogen Corporation, Carisbad, CA, USA), for sample homogenization. The final mixture was divided into 1 mL aliquots and stored in a freezer at −80° for further analysis.

### 2.6. Isolation of RNA from Leucocytes and Preparation of cDNA for Real-Time Polymerase Chain Reaction (RT-PCR)

Total RNA was prepared following Trizol^®^ reagent instructions. The total RNA extraction was quantified using a Gene Quant^®^ (Pharmacia Biotech, Piscataway, NJ, USA) spectrophotometer, considering the 260/280 ratio. The ratios found in the samples were close to 1.8 and considered optimal for use.

For the production of complementary DNA (cDNA), we used the High-Capacity cDNA Reverse Transcription Kit (Applied Biosystems, Foster city, CA, USA). The final cDNA concentration was 30 ng/µL (20 µL totals). The cDNA was diluted to the concentration required for the efficient amplification of each (target and endogenous) gene.

### 2.7. RT-PCR for CHRNA7, CHRFAM7A, and TNF-α

The RT-PCR reactions were performed using the TaqMan system (Applied Biosystems), which consists of a pair of primers and a probe labeled with a fluorophore. The human Glyceraldehyde-3-phosphate dehydrogenase (GAPDH) gene (Applied Biosystems) was chosen as the endogenous control of the reaction, which serves to normalize the expression of the target gene in the different samples. The probe contained in the primer GAPDH is labeled with the VIC fluorophore, while the primers for the targets are labeled with the FAM fluorophore.

PCR reactions contained the following amounts of the reagents: 40 ng cDNA, 5 µL Master Mix, 0.25 µL target primer, 0.25 µL endogenous primer, and ultrapure water, totaling 10 µL final. PCR reactions were analyzed on Step one Plus^®^ equipment (Applied Biosystems, Foster City, CA, USA) and the analyses for each sample occurred in duplicate. We used the following universal cycling temperatures: 95 °C for 3-min and 45 cycles of 95 °C for 5 s and 60 °C for 20 s. The primers used for mRNA quantification of the analyzed genes were for human: *GAPDH*, 4326317E; *TNF-α*, Hs00174128_m1; *CHRNA7*, Hs01063372_m1 and *CHRFAM7A*, Hs.PT.58.20622816.

### 2.8. Acute Physical Exercise Sessions Protocols

The acute physical exercise sessions (AE, RE and CE) lasted 50 min. In the RE session, 3 sets of 12 RM was performed with 1-min rest between sets for 9 exercises (chest press, leg press 45º, high pull, leg flexion, knee extension, triceps pulley, barbell, calf press and abdominal crunch) [30]. The AE session consisted of walking or running for 5 min at 40–45% of VO_2peak_, 20 min at 45–50% of VO_2peak_, 20 min at 50–55% of VO_2peak_ and 5 min between 40–45% VO_2peak_ [30]. The CE session consisted of RE followed by AE. Total of 7 exercises were performed (the same as RE, without calf press on leg press 45° and abdominal crunch) with 2 sets of 12 RM and 1-min rest between sets. After that, the participants walked or ran for 5 min at 40–45% of VO_2peak_, 7.5 min at 45–50% of VO_2peak_, 7.5 min at 50–55% of VO_2peak_ and 5 min between 40–45% VO_2peak_ [30].

The intensities of the acute physical exercise sessions followed the recommendations of the American College of Sports and Medicine for individuals with T2DM [30]. The load for RE was defined using the 12 RM test (45–55% of 1 repetition maximum) and the intensity of AE was calculated based on the VO_2peak_ result achieved in the cardiorespiratory test. A summary of the exercise sessions is in Table 1. The control of the acute physical exercise sessions was carried out through the rating of the perceived exertion scale (6–20) and through the monitoring of the heart rate (Polar FT1, Kempele, Finland) (participants should be between 12–14 on the exertion scale and between 60–70% of maximum heart rate) [31].

### 2.9. Statistical Analyses

Initially, data distribution was analyzed using the Shapiro–Wilk test. Logarithmic transformation was applied to non-normally distributed data: body mass, waist circumference, A1c, homeostasis model assessment of insulin resistance (HOMA-IR), upper-body strength (after the transformation, the data of these variables become normal). This study has two research objectives; the first one was to verify the existence of baseline differences between HNW and OT2D groups, thus comparisons between groups were performed through an unpaired Student’s t-test for all variables, except for *CHRNA7*, due to the low occurrence of expression among participants.

The second objective was to analyze the effects of AE, RE and CE on *CHFAM7A* and *TNF-α* expression of the HNW and OT2D groups. As there was a baseline difference between the groups of subjects, we used an analysis of co-variance (ANCOVA) to compare the differences between the groups (subjects) and the time (pre and post each type of exercise session). In addition, we compared the differences between each exercise session within each group of subjects, using a two-way analysis of variance (ANOVA) with repeated measures considering the interaction between the test time (pre- and post-exercise) and type of exercise (AE, RE and CE). As only the time effect was observed with CHRFAM7A in the two-way ANOVA, we compared the isolated effect of each exercise session in each of the groups of subjects with a paired t-test. We also compared the percentage deltas of changes (Δ%) of each group/exercise evaluation (the different subjects and each exercise session, and even the groups of subjects and different exercise sessions) using unpaired Student’s t-test. We also calculated the d-effect size for the baseline results of CHRN7A, CHRFAM7A and TNF, and for pre–post (AE, RE and CE) CHRFAM7A from HNW and OT2D (main results of the present work). Finally, the Pearson correlation analyses were applied to test associations between baseline gene expressions. The software package used was STATISTICA 6.0 (StatSoft, Inc., Tulsa, OK, USA). The level of statistical significance was set at *p* < 0.05. Data are presented as mean ± standard deviation (SD).

## 3. Results

### 3.1. Group Characterization

Table 2 shows the significant difference in body composition variables (weight, BMI, waist circumference, fat-free mass, body fat) between groups (*p* = 0.01).

The OT2D also presented higher values of insulin, glucose, glycated hemoglobin, HOMA-IR (*p* < 0.001 for all variables) and SBP and DBP (*p* = 0.01) when compared to the HNW. In addition, the OT2D had lower high-density lipoprotein cholesterol levels (*p* = 0.05).

In relation to physical fitness, the HNW had higher VO_2peak_ than OT2D (*p* = 0.00) and there was no significant difference for the upper (*p* = 0.21) and lower limb (*p* = 0.66) muscle strength (Table 2).

### 3.2. Baseline Gene Expression (Pre Exercise Sessions)

Basal gene expression was established in pre-exercise sessions. The *CHRNA7* expression occurred late between the 38th and 41st cycle, and there was a large variation in its expression in both groups. No statistical differences were observed between the participants who expressed *CHRNA7* in both groups (*p* = 0.58) (Figure 3A). However, it is noteworthy that while 49% of the HNW baseline samples expressed this gene (19 of 39 samples), only 24% of the OT2D baseline samples expressed it (11 of 45 samples).

In contrast, all participants from both groups consistently expressed *CHRFAM7A* and *TNF-α*. Comparing all baselines from all exercise sessions together, the OT2D presented higher *CHRFAM7A* (*p* = 0.01) and *TNF-α* (*p* = 0.001) expression when compared to the HNW (Figure 3B,C). These results are similar when comparing each exercise session separately (*CHRFAM7A:* AE *p* = 0.01; RE *p* = 0.04; CE *p* = 0.04; and *TNF-α*: AE *p* < 0.0001; RE *p* = 0.01; CE *p* = 0.02). Additionally, there was a positive correlation between the CHRFAM7A and the TNF-α expression in OT2D (r = 0.75, *p* = 0.01).

### 3.3. Response to Physical Exercise Sessions

The response of CHRNA7 on the acute physical exercise (AE, RE and CE), as well as its basal expression, presented high within and between the subject’s variability in HNW and OT2D groups. Regarding the between-subjects variation, there were increases, reductions or undetectable gene expression levels from pre to post the exercise sessions (AE, RE and CE). Thus, no statistical differences were observed for any exercise session and group among the participants who had the CHRNA7 expression in at least one of the analysis moments (pre–post). Figure 4 demonstrates an example of *CHRNA7* behavior on the AE session in OT2D.

There were no group/time differences in CHRFAM7A and TNF- α gene expression (both as in the comparison using ANCOVA, as for the Δ%). Comparing exercise sessions within the same group of subjects, we observed an ANOVA time effect for CHRFAM7A gene expression for both HNW (*p* = 0.01) and OT2DM (*p* < 0.001) (Figure 5A,B). Finally, isolating time effects within each exercise session using a paired t-test, no significant changes were observed for TNF-α gene expression in any of the groups. However, there was an increase in the CHRFAM7A gene expression in AE (*p* = 0.04), RE (*p* = 0.04) and CE (*p* = 0.04) for HNW, and in AE (*p* = 0.01) and CE (*p* = 0.01) for OT2D (Figure 5A,B). No significant changes were observed in RE for OT2D (*p* = 0.43) (Figure 5A).

## 4. Discussion

We hypothesized that in the baseline, OT2D would have lower *CHRNA7* expression with higher *CHRFAM7A* and *TNF-α* expression in leukocytes when compared with HNW. Although there are no statistical differences between the groups in CHRNA7A, we observed a different pattern of expression response: while 49% (19 from 39 baseline samples) of the HNW expressed *CHRNA7*, only 24% (11 from 45 baseline samples) of the OT2D demonstrated detectable levels. On the other hand, as hypothesized the OT2D demonstrated significantly higher *CHRFAM7A* and *TNF-α* basal expression when compared to the HNW. Due to physiological protective factor of CHRFAM7A, as observed in certain diseases and acute stressful conditions, the secondary hypothesis of the present study was the increase in CHRFAM7A expression after exercise sessions, especially in HNW. In addition, these exercise effects could vary depending on your type. As expected, we observed an increase in the expression of CHRFAM7A in all types of exercise in HNW, and in AE and CE sessions in OT2D. Except for the non-increase in CHRFAM7A expression in RE session practice by OT2D, no other differences were observed between the groups in the exercise sessions.

Similar to our findings, a previous study already showed the *CHRNA7* expression has been highly variable or undetectable in human leucocytes, while CHRFFAM7A expression seems to be consistently expressed among individuals [11]. A possible explanation for these variability of CHRNA7 expression responses is the type of primer used to verify its expression, because the first primers used did not differentiate the *CHRNA7* from its partial duplication (*CHRFAM7A*), since they identified only the common oligonucleotides between the genes [11,13]. In the present study, the use of primers to identify specific oligonucleotides for each of the genes studied, preventing errors in the identification and differentiation between them [32,33].

In addition, there is evidence in the literature for alterations in transcriptional mechanisms of the *CHRNA7* expression such as as polymorphism or deletions, which could be involved with the deficiency in the *α*7nAChR expression and consequent pathogenesis of neurodevelopmental disorder or predisposition to other diseases [34,35,36]. Cedillo et al. (2015) reported high levels of α7nAChR expression in the leukocytes of patients with sepsis, and those who demonstrated greater receptor expression had a better prognosis of the disease [34]. Thus, another possibility for the *CHRNA7* baseline expression reduced in OT2D group could be an alteration in transcriptional mechanisms predisposing these individuals to the highest of inflammatory levels and development of associated diseases.

In this present study, the evidence points to deregulation in the *CHRFAM7A* and a *CHRNA7* expression baseline. Studies in vitro with the immune cells show that when these cells were stimulated using lipopolysaccharide, there was a *CHRFAM7A* expression down regulation (pro-inflammatory) and a *CHRNA7* expression up regulation (anti-inflammatory), in an attempt to control acute inflammation [13,14]. However, in patients with chronic inflammatory bowel diseases, Baird et al. (2016) observed a *CHRFAM7A* expression up regulation and a *CHRNA7* expression down regulation, as in the present study [18]. In both studies, the sample used was non-neuronal cells (enterocytes *x* leucocytes) and participants underwent chronic inflammation (overweight/obese individuals with T2DM *x* inflammatory bowel disease) [12,13,18]. Thus, in the chronic inflammation context, seems there is a baseline disruption this control mechanism of inflammation: a *CHRFAM7A* up regulation and a *CHRNA7* down regulation contributing to the increase in the inflammatory levels, as evidenced in this study by the *TNF-α* expression increase. The baseline correlation between the *CHRFAM7A* and the *TNF-α* expression strengthens this belief.

So, in the OT2D group, smaller number of participants presented *CHRNA7* expression when compared with the HNW, and this reduction might be associated with the *CHRFAM7A* expression increase, and, consequently, increased *TNF-α* expression. In that way, it is possible that the *CHRNA7* expression reduction and consequent CAP activity reduction observed in obese people associated with T2DM may contribute to the kind of chronic inflammation usually observed in these individuals. Accordingly, *CHRNA7* expression and the CAP activation increase would contribute to the reduction of systemic inflammation in this population and could become a new therapeutic target in the treatment of obesity and T2DM.

Based on recent findings showing an increased *CHRFAM7A* expression under conditions of acute stress induced by an acute disease as a protective factor [16,17,18], we hypothesized that acutely, physical exercise could also increase the expression of this gene, especially under conditions of oxidative stress, such as aerobic exercise, and in greater magnitudes for the OT2D group. For this, the effect of three different protocols of physical exercise (AE, RE and CE) on the *CHRNA7*, *CHRFAM7A* and *TNF-α* expression were investigated. As a baseline, the *CHRNA7* expression showed both a within/between-subjects’ variable response, and it was not possible to verify a response pattern for this gene. In addition, there was not significant alteration in *TNF-α* expression for all acute physical exercise, while for *CHRFAM7A* there was a time effect for the increase in expression of this gene in both groups. In addition, in the isolated analysis of each exercise session, we observed an increase in CHRFAM7A expression in the three sessions performed by HNW and in the AE and CE realized by OT2D.

This was the first study that investigated the effect of a physical exercise session on *CHRNA7* and *CHRFAM7A* expression. The results showed that a single physical exercise session seems to be able to change the *CHRFAM7A* expression, as well in other conditions of acute stress, such as certain acute illnesses. These results show that the increase in the acute expression of CHRFAM7A must be a physiological protection factor against acute and inflammatory stressors and the presence of obesity and DM2 does not alter the response to acute stress stimulus. Although the inflammatory profile is known to influence fatigue status [37] and the OT2D group had lower baseline cardiorespiratory fitness here, all the participants reached the individual exercise intensity prescribed (by HR and RPE), and the three exercise protocols challenging cardiorespiratory fitness are what led to the increases in CHRFAM7A expression in both groups.

Probably, the increase in oxidative stress is the main factor responsible for this change, since regardless of the group analyzed, in all sessions where the aerobic stimulus was present (AE and CE), there was an increase in the expression of *CHRFAM7A.* Interestingly, acute diseases and conditions that generate high oxidative stress and reactive oxygen species (COVID-19, brain ischemia, lung injury, LPS exposure and skin burn) exert similar effects [16,17,18,38].

The only exercise session that was not observed to increase the expression of *CHRFAM7A* was RE in OT2D. In addition to the issue of oxidative stress, another mechanism that may be involved is the increase in the acute production of IL-6 through muscle contraction promoting a more anti-inflammatory environment [20,21]. The RE session also may have increased IL-6 levels, since in this type of exercise, the main muscle groups of the upper and lower limbs were exercised, while in AE the focus was greater on the lower limbs due to the type of exercise chosen (walk).

This study had some limitations. We found small to moderate effects sizes for group difference at baseline and effect of exercise on *CHRNA7* and *CHRFAM7A,* however, small changes in gene expression can induce significant effects when frequently repeated [39,40]. Another limitation is the high risk of a type one error in our analysis with low statistical power sample size. Although the general time effect of exercise increasing CHRFAM7A expression was robust using the two-way ANOVA analysis, the comparison between types of exercise must be interpreted with caution, since no interaction was observed in the two-way ANOVA and those exploratory comparisons were obtained using simple t-tests. Finally, we did not test the participants’ eating and sleeping habits preceding each session, which could have influenced our results. On the other hand, we gave to the participants a standardized breakfast meal to eat before exercise sessions, and participants were recommended to maintain the same eating and sleeping pattern across the nights previous to exercise.

In conclusion, OT2D presented a higher TNF-α and CHRFAM7A baseline expression, in addition to evidence of decreased CHRNA7A expression in leukocytes when compared with HNW. Thus, low-grade inflammation usually observed in obesity and associated with T2DM may be related to the reduction of CAP activity. On the other hand, acute physical exercise induces increased CHRFAM7A expression, especially when the aerobic component is present. This acute increase in the expression of CHRFAM7A must be a physiological protective factor, as observed in certain diseases and acute stressful conditions.

## Figures and Tables

**Figure 1 biomedicines-11-00565-f001:**
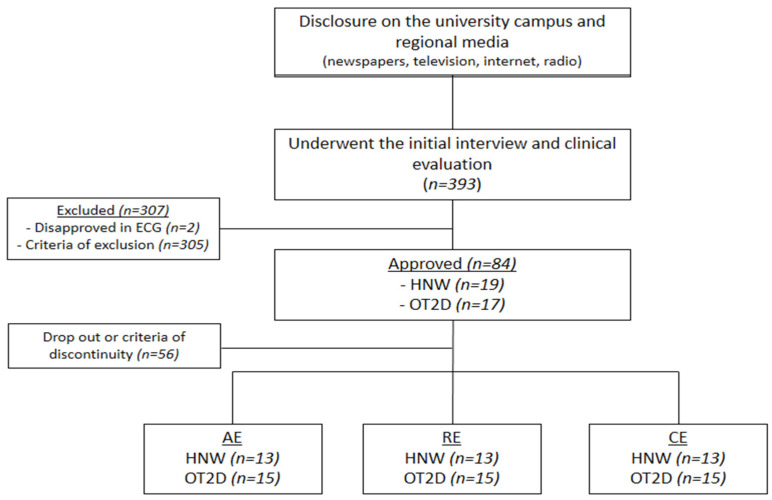
**Flowchart.** ECG = electrocardiogram; HWM = healthy normal weight individuals; OT2D = overweight individuals with T2DM.

**Figure 2 biomedicines-11-00565-f002:**
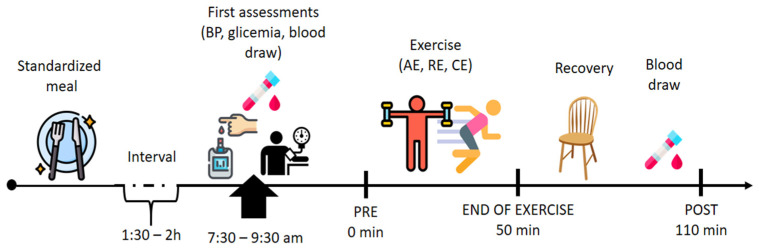
**Experimental design.** AE = aerobic exercise; RE = resistance exercise; CE = combined exercise. In summary, the participants arrived at the laboratory between 7:30 a.m. and 9:30 a.m. Before starting the session, systolic (SBP) and diastolic (DBP) blood pressure were measured. Immediately after, blood glucose measurement was performed using a glucometer device, (only for the OT2D group with the aim of maintenance of similar baseline glycemic levels between physical exercise sessions). Then blood collection was performed before the beginning of the session and after 110 min post exercise session.

**Figure 3 biomedicines-11-00565-f003:**
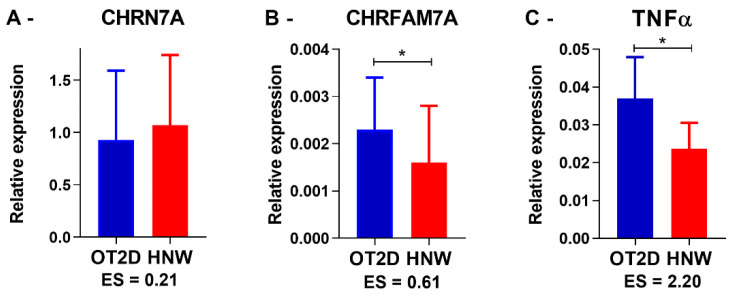
**Baseline gene expression.** OT2D = overweight individuals with T2DM. HNW = healthy normal weight individuals. CHRN7A n = 11 in OT2D and 19 in HNW. *CHRFAM7A* and *TNF-α n =* 45 in OT2D and 39 in HNW. * = *p* < 0.05 (Student’s independent *t*-test). ES = Effect size value.

**Figure 4 biomedicines-11-00565-f004:**
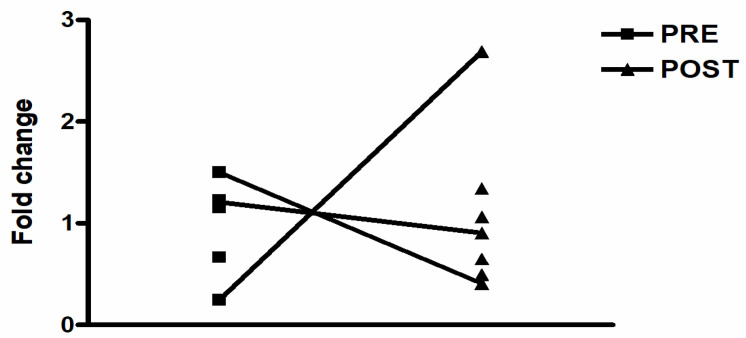
Example of *CHRNA7* behavior on the aerobic exercise session in OT2D.

**Figure 5 biomedicines-11-00565-f005:**
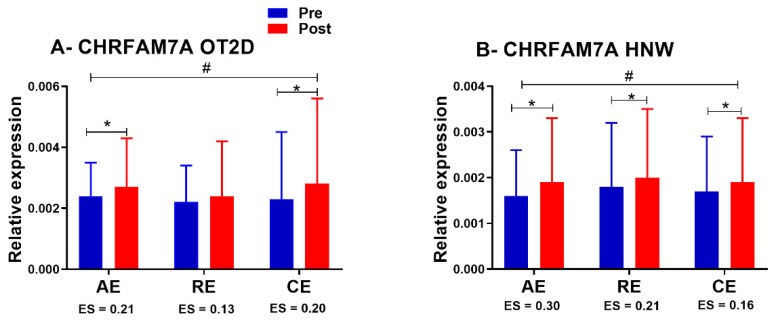
**CHRFAM7A expression in exercise sessions.** OT2D = overweight individuals with T2DM. HNW= healthy normal weight individuals. # = *p* < 0.05 (time effect in ANOVA). * *p* < 0.05 (Student’s dependent *t*-test). ES = Effect size value.

**Table 1 biomedicines-11-00565-t001:** Description of exercise sessions.

Exercise Session	RE	CE	AE
**Strength exercise**			
Leg press	3 × 12	2 × 12	-
Knee extension	3 × 12	2 × 12	-
Knee flexion	3 × 12	2 × 12	-
Chest press	3 × 12	2 × 12	-
High pull	3 × 12	2 × 12	-
Barbell	3 × 12	2 × 12	-
Triceps pulley	3 × 12	2 × 12	-
Calf press	3 × 12	-	-
Abdominal crunch	3 × 12	-	-
Rest (s) interval between sets (sec.)	60	60	-
Intensity (% of 1 repetition maximum)	45–55	45–55	-
Total duration strength training (min.)	50	25	-
**Aerobic exercise**			
40–45% VO_2_max (min.)	-	5	5
45–50% VO_2_max (min.)	-	7.5	20
50–55% VO_2_max (min.)	-	7.5	20
40–45% VO_2_max (min.)	-	5	5
Total duration aerobic training (min.)	-	25	50

RE = resistance exercise session; CE = combined exercise session; AE = aerobic exercise session.

**Table 2 biomedicines-11-00565-t002:** Baseline characteristics of groups.

	HNW (n = 13)	OT2DM (n = 15)
Sex (F/M)	6/7	7/8
Age (years)	50.00 ± 6.20	52.00 ± 4.14
Diagnosis of T2DM (years)	-	6.27 ± 5.89
**Body Composition**		
Weight (kg)	69.48 ± 10.48 *	89.12 ± 9.61
Body mass index (kg/m^2^)	24.42 ± 1.87 *	30.45 ± 3.34
Waist circumference (cm)	86.15 ± 7.64 *	10,167 ± 7.32
Fat free mass (kg)	45.09 ± 14.65 *	55.67 ± 11.20
Body fat (%)	29.94 ± 8.94 *	37.09 ± 8.00
**Biochemical markers**		
Insulin (µU/mL)	7.87 ± 3.62 *	12.32 ± 4.02
Fasting glycaemia (mg/dL)	83.33 ± 9.94 *	158.40 ± 61.90
A1C (%)	5.60 ± 0.31 *	7.83 ± 2.44
HOMA-IR	1.65 ± 0.84 *	4.72 ± 2.31
LDL cholesterol (mg/dL)	139.58 ± 36.23	120.20 ± 39.58
HDL cholesterol (mg/dL)	48.75 ± 11.14 *	40.93 ± 8.96
Triglycerides (mg/dL)	119.83 ± 41.58	160.00 ± 69.46
**Hemodynamics**		
SBP (mmHg)	107.17 ± 10.63 *	127.45 ± 8.39
DBP (mmHg)	77.48 ± 4.24 *	88.15 ± 4.37
**Physical fitness**		
VO_2peak_ (kg/m^2^)	27.56 ± 4.68 *	22.11 ± 4.77
12RM Leg press 45° (kg)	50.83 ± 22.81	55.33 ± 29.85
12RM chest press (kg)	8.80 ± 6.49	11.79 ± 7.10

OT2DM—obese/overweight com type 2 diabetes mellitus; CTR—eutrophic control; AIC—glycated hemoglobin; LDL Cholesterol—low density lipoprotein cholesterol; HDL cholesterol—high density lipoprotein cholesterol; A1C—glycated hemoglobin; HOMA-IR—homeostasis model assessment—insulin resistance; SBP—systolic blood pressure; DBP—diastolic blood pressure; VO_2peack_—peak oxygen consumption; 12RM—twelve repetitions maximum. Values are media ± standard deviation. * Significantly different between groups (*p* < 0.05).

## Data Availability

The raw data supporting the conclusions of this manuscript will be made available by the authors, without undue reservation, to any qualified researcher.

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
