# Peer review of "Effects of Different Exercise Types on Chrna7 and Chrfam7a Expression in Healthy Normal Weight and Overweight Type 2 Diabetic Adults"

_biomedicines, 2023, doi:10.3390/biomedicines11020565_

Round 1

Reviewer 1 Report

Comments to the Authors of manuscript number: biomedicines-2153919 entitled “EFFECTS OF DIFFERENT EXERCISE TYPES ON CHRNA7 AND CHRFAM7A EXPRESSION IN NORMAL WEIGHT AND OVERWEIGHT TYPE 2 DIABETES ADULTS”.

1. Abstract, purpose – it is not clear what is the relation between chronic inflammation and exercise. It should be presented by other way

2. some expressions do not have to be capitalized. It should be corrected along the text (e.g. the 3rd paragraph of the introduction)

3. the same as above (the Cholinergic Receptor, Nicotinic, Alpha 7 Family With Sequence Similarity 7A)

4. The introduction presents the problem very well, and the hypothesis and the goal are well written

5. Excluded criteria are well described.

6. Finally, there were 13 healthy and 15 overweight patients with T2DM, but there is no information about the age and gender

7. not “basel blood” but basal

8. what is 12 RM? It should be explained

9. Table 1. Some data should be shifted into the part of material and methods, where the patients are described

10. reference for HOMA-IR

Author Response

Authors’ response to reviewers

First, we thank the reviewers for spending their time in the evaluation of our manuscript. We really appreciate this volunteer effort. We have revised our manuscript with all changes marked in yellow and our detailed response to each comment is described below. 

We are looking forward to your response.

Yours sincerely

Reviewer 1

Comments to the Authors of manuscript number: biomedicines-2153919 entitled “EFFECTS OF DIFFERENT EXERCISE TYPES ON CHRNA7 AND CHRFAM7A EXPRESSION IN NORMAL WEIGHT AND OVERWEIGHT TYPE 2 DIABETES ADULTS”.

  1. Abstract, purpose – it is not clear what is the relation between chronic inflammation and exercise. It should be presented by other way

Response: We added a sentence to better explain this relation (page 1).

  1. some expressions do not have to be capitalized. It should be corrected along the text (e.g. the 3rdparagraph of the introduction)

Response: Thank you for catching this error, we reviewed these terms (page 2).

  1. the same as above (the Cholinergic Receptor, Nicotinic, Alpha 7 Family With Sequence Similarity 7A)

Response: Thank you for catching this error, we reviewed these terms (page 2).

  1. The introduction presents the problem very well, and the hypothesis and the goal are well written

Response: Thank you.

  1. Excluded criteria are well described .

Response: Thank you.

  1. Finally, there were 13 healthy and 15 overweight patients with T2DM, but there is no information about the age and gender

Response: We added this missing information (page 3).

  1. not “basel blood” but basal

Response: We correct this error (page 4).

  1. what is 12 RM? It should be explained

Response: Thank you for this observation, we added more information for this abbreviation (page 4).

  1. Table 1. Some data should be shifted into the part of material and methods, where the patients are described

Response: We revised the text and table to keep inclusion criteria information in the methods and the descriptive statistics about the selected participants in the results. We also took this opportunity to improve the table design (page 8).  

  1. reference for HOMA-IR

Response: We added a reference for it (page 5).

Reviewer 2 Report

This is a very well designed study that needs some work on the presentation. Below are my comments

Introduction

1. In the introduction you should start out by providing epidemiological evidence of why obesity is an issue before getting into how obesity can increase risks for chronic diseases such as Type 2 Diabetes. 

2. There are several run-on sentences in your introduction. Please have someone proof read the introduction to improve writing quality. Also try to use phrases such as "such as" instead of "like"

Methodology

1. How were subjects recruited? 

2. At the beginning of the methodology you state that there are 3 groups, healthy normal weight, obese and obese with T2DM however, later in your methodology you only have HNW and OT2DM. Why is that?

3. You seem to do a great job controlling for nutrition. Did you also control for prior night's sleep? There is significant evidence that sleep deprivation influences inflammatory response

4. The figure on page 5 is extremely helpful. Please make sure you label it.

5. The description of the exercises is excellent. I would recommend some sort of figure to help readers visualize it. 

6. You state that you controlled the exercise intensity using RPE and HR. Please elaborate what RPE you wanted them to be at during exercise, same with HR

7. How did you reduce risks for Type I error? With that many tests there is a likelihood for a Type I error.

8. When you transformed the variables were they able to be transformed?

9. Did you complete an a priori power analysis?

10. Why not run a 1 between (2 groups) by 2 within (3 exercise types, 2 time points), mixed ANOVA with standardized scores? That would be the ideal analysis as splitting it up into multiple analyses increased the risk for Type I error. After that if there are signficant differences you can run post-hocs, depending on what outcome is significantly different. This would also reduce the risk for Type I errors that may occur when you calculate delta change.

11. Did you calculate effect size? Considering the complexity of the analysis i would recommend calculating the generalized effect size and presenting that

12. Did you calculate post-hoc power?

13. What did you use to correct for the multiple post-hoc analyses? 

Results

The results are well presented. Although this is just a personal preference, considering you are submitting your work to an open access journal, I'd recommend adding color to your figures. It will really help the figures stand out.

Discussion

Although I don't think your results will change much when you perform a 1 between x 2 within mixed ANOVA using standardized results/an ANCOVA I do think that there may be some minor changes in your results. Further, I don't know the study was properly powered and the effect sizes were large enough to warrant a significant clinical effect. I will reserve further comment on the discussion section after you have completed those statistical analyses.

There are several limitations to the study, however, you have only identified one. Based on what I have read in the methodology, not controlling for prior night's sleep and sleep quality could significantly impact your results. Further, I would recommend work that examines how baseline inflammation impact fatigue levels, which may have contributed to some of your results (especially the results presented in Figure 4). 

Author Response

Authors’ response to reviewers

First, we thank the reviewers for spending their time in the evaluation of our manuscript. We really appreciate this volunteer effort. We have revised our manuscript with all changes marked in yellow and our detailed response to each comment is described below. 

We are looking forward to your response.

Yours sincerely

Reviewer 2

This is a very well designed study that needs some work on the presentation. Below are my comments

Introduction

  1. In the introduction you should start out by providing epidemiological evidence of why obesity is an issue before getting into how obesity can increase risks for chronic diseases such as Type 2 Diabetes. 

Response: We added more information about the epidemiology of obesity and why it is an issue (page 1, first paragraph of Introduction) (page 1).

  1. There are several run-on sentences in your introduction. Please have someone proof read the introduction to improve writing quality. Also try to use phrases such as "such as" instead of "like"

Response: Thank you for pointing this out, we reviewed these issues in the text (page 1 and 2).

Methodology

  1. How were subjects recruited? 

Response: This information was presented in figure 1 (flowchart), and since it was not very clear we now added more details about the participants recruitment in the text (page 3).

  1. At the beginning of the methodology you state that there are 3 groups, healthy normal weight, obese and obese with T2DM however, later in your methodology you only have HNW and OT2DM. Why is that?

Response: The first sentence of the methodology where the information cited by the present reviewer is located refers to the main study trial in which the present work is a secondary arm. For the purpose of the present study, only OT2D and HNW individuals were analyzed so we added a sentence to clarify this information (page 2).

  1. You seem to do a great job controlling for nutrition. Did you also control for prior night's sleep? There is significant evidence that sleep deprivation influences inflammatory response

Response: The sleep was not controlled, although we recommended the participants to maintain their normal sleeping habits. We added this information in the methods (page 4).

  1. The figure on page 5 is extremely helpful. Please make sure you label it.

Response: Thank you for this observation, we modified the Figure 2 and we added more information in the legend to improve clarity (page 5)

  1. The description of the exercises is excellent. I would recommend some sort of figure to help readers visualize it. 

Response: We added a table to simplify the understanding of the exercise prescription. Please, let us know if you believe a figure would still be better than the table (page 7).

  1. You state that you controlled the exercise intensity using RPE and HR. Please elaborate what RPE you wanted them to be at during exercise, same with HR

Response: We detailed this information in the methods section (page 7).

  1. How did you reduce risks for Type I error? With that many tests there is a likelihood for a Type I errorr a Type I error.

Response: Initially we did not run multiple tests, but rather we ran a two-way analysis of variance (ANOVA) with repeated measures considering interaction between test time (pre- and post-exercise) and type of exercise (AE, RE, and CE) that confirmed the time-effects for CHRFAM7A. In this way we can prevent type 1 error. Next, we only explored the the time-effects observed in the 2 way ANOVA in each of the 3 exercise types, by T-tests for this variable (CHRFAM7A). We explored the time-effect within each exercise session by dependent T-tests because we could not show post-hoc results for the ANOVA, since there was no significant group*time interaction. Please, see more information on this issue in the answer to topic 10 (below).

  1. When you transformed the variables were they able to be transformed?

Response: After the transformation, these data became normally distributed. We added this information in the text (page 7).

  1. Did you complete an a priori power analysis?

Response: As mentioned, this study was part of a larger trial and the a priori power calculation were done for the main variables (Bonfante, Duft et al. 2021, Bonfante, Monfort-Pires et al. 2022). Therefore, we just presented the a posteriori power calculation for baseline genes and pre-post expression CHRFAM7A in all exercise conditions. The power was calculated using G*Power 3.2.1 software, based on genes expression of the present study, and  Using a study design of a f test and/or t test with p < 0.05 and 95% power. The power was calculated using G*Power 3.2.1 software, using a study design of an f test and/or t test with p < 0.05 and 95% power. The baseline power results were: CHRN7A t = 0.13; CHRFAM7A t = 0.47; TNF-α t = 0.99. In the exercise sessions, OT2D showed a CHRFAM7A power of: AE f = 0.42, t = 0.20; RE f = 0.18, t = 0.12; CE f = 0.38, t = 0.19, while HNW showed a power of AE f = 0.67, t = 0.29; RE f = 0.36, t = 0.19; CE f = 0.22, t = 0.15).We have inserted this information into the text of methodology and discussion/limitations (page 3 and 12).

Bonfante, I. L. P., et al. (2021). "Acute/Chronic Responses of Combined Training on Serum Pro-thermogenic/Anti-inflammatory Inducers and Its Relation With Fed and Fasting State in Overweight Type 2 Diabetic Individuals." Front Physiol 12: 736244.

Bonfante, I. L. P., et al. (2022). "Combined training increases thermogenic fat activity in patients with overweight and type 2 diabetes." Int J Obes (Lond) 46(6): 1145-1154.

  1. Why not run a 1 between (2 groups) by 2 within (3 exercise types, 2 time points), mixed ANOVA with standardized scores? That would be the ideal analysis as splitting it up into multiple analyses increased the risk for Type I error. After that if there are significant differences you can run post-hocs, depending on what outcome is significantly different. This would also reduce the risk for Type I errors that may occur when you calculate delta change.

Response: In the comparison between groups for exercise response we applied ANOVA two-way on the CHRN7A gene and due to the differences at baseline between the groups on the CHRFAM7A and TNF genes we applied ANCOVA. As a result, we did not observe differences between the groups. These two tests avoid the type 1 error cited by the reviewer in comparisons between groups versus exercise responses. Due to the lack of difference between groups in these analyzes, we did not apply other tests with less power of analysis and that could induce a type 1 error.

In addition, comparing the effects of each exercise session within each group (HNW and OT2D), we also applied two-way ANOVA (to avoid type 1 error). We did not observe a group versus time effect in these analyses, which precludes the application of post hoc tests. However, as we observed in these analyzes of CHRFAM7A a clear effect of time for both groups (which shows a significant effect of exercises on this gene). As a result, we decided to apply the T-test (less robust and more subject to type 1 error) to try to better explore the responses of each type of exercise, however, we recognize that this is not the best test.

Thus, we inserted this as a limitation in the discussion, focusing on the fact that exercise in general (regardless of the type) has an influence on the expression of CHRFAM7A (due ANOVA time effect in both, HNW and OT2D groups), but that the differences observed between the types of exercises should be carefully observed due to the T-test applied (page 12).

  1. Did you calculate effect size? Considering the complexity of the analysis i would recommend calculating the generalized effect size and presenting that

Response: Thank you for this suggestion. We agree that effects size could bring valuable information to interpret the data and thus we added the pre-post effect size for: baseline CHRN7A, CHRFAM7A and TNF; and in Pre-post (AE, RE and CE) CHRFAM7A of HNW and OT2D (figures 3 and 5). We have also inserted comments on this point in the discussion/limitation (page 12).

  1. Did you calculate post-hoc power?

Response: Yes, we added this information in the text and in question 9 for the reviewer answer.

  1. What did you use to correct for the multiple post-hoc analyses? 

Response: We did not present any post-hoc since we did not find any interaction between time and type of exercise.

Results

The results are well presented. Although this is just a personal preference, considering you are submitting your work to an open access journal, I'd recommend adding color to your figures. It will really help the figures stand out.

Response: Thank you for your suggestion, we replaced most of the black and white figures to color figures and we agree they look much better now.

Discussion

Although I don't think your results will change much when you perform a 1 between x 2 within mixed ANOVA using standardized results/an ANCOVA I do think that there may be some minor changes in your results. Further, I don't know the study was properly powered and the effect sizes were large enough to warrant a significant clinical effect. I will reserve further comment on the discussion section after you have completed those statistical analyses.

There are several limitations to the study, however, you have only identified one. Based on what I have read in the methodology, not controlling for prior night's sleep and sleep quality could significantly impact your results. Further, I would recommend work that examines how baseline inflammation impact fatigue levels, which may have contributed to some of your results (especially the results presented in Figure 4). 

Response: We agree that those limitations are important and therefore we described it more clearly in the limitations section and we add other points in discussion. In addition, we inserted a reference about the baseline inflammation effects on fatigue and we discuss on discussion topic; however possible higher inflammation in the OT2D does not seems to have influenced the CHRFAM7A expression in exercise session (except in RE) ( 11 and 12)

Round 2

Reviewer 2 Report

I appreciate the authors addressing all of my concerns. The Table was a much better idea than the Figure.